biophysics/physiological optics

spontaneous blinking, corneal reflex, video oculography, eye movements, exponentially modified Gaussian function

**Author for correspondence:**
Julián Espinosa
e-mail: julian.espinosa@ua.es

# Comparative analysis of spontaneous blinking and the corneal reflex

Julián Espinosa[1,2], Jorge Pérez[1,2] and David Mas[1,2]

[1]Department of Optics, Pharmacology and Anatomy, University of Alicante, Alicante, Spain
[2]University Institute of Physics Applied to Sciences and Technologies, University of Alicante, Alicante, Spain

JE, 0000-0001-6817-3117

Ocular surface health, the cognitive status, psychological health or human neurological disorders, among others, can be assessed by studying eye blinking, which can be differentiated in spontaneous, reflex and voluntary. Its diagnostic potential has provided a great number of works that evaluate their characteristics and variations depending on the subject's condition (sex, tiredness, health, …). The objective of this study was to analyse the differences in blinking kinematics of spontaneous and reflex blinks, distinguishing between direct and consensual reflexes, using a self-developed, non-invasive and image processing-based method. A video-oculography system is proposed using an air jet driven by a syringe to induce reflex and a high-speed camera to record the blinking of both eyes. The light intensity diffused by the eye changes during blinking and peaks when the eyelid closes. Sixty-second sequences were recorded of 25 subjects blinking. Intensity curves were off-line fitted to an exponentially modified Gaussian (EMG) function, whose $\sigma$, $\mu$ and $\tau$ parameters were analysed. A two-way analysis of variance (ANOVA) of these parameters was conducted to test the influence of the subject, the eye and blink type. In the closing phase, direct and consensual corneal reflexes are faster than spontaneous blinking, but there was no significant difference between them, nor between right and left eyes. In the opening phase, the direct corneal reflex was the slowest and significant differences appeared between right and left eyes.

## 1. Introduction

A blink is a temporary closure of both eyes which involves movements of the upper and lower eyelids [1] to keep the eye hydrated by distributing tear film over the entire eye surface [2], and to protect it from foreign objects [3]. Eyelid movements require simple neural commands and a few active forces, so blinking represents a normal, simply observable and easily

accessible phenomenon that reflects central nervous system activation processes without voluntary manipulation. Its analysis can reveal some muscular or neural disorders [3–5], which makes blinking a highly relevant source of information and encourages the examination of its characterizing parameters.

Three blink types can be differentiated according to the subject's will: spontaneous, voluntary and reflex blinking. Spontaneous blinking occurs regularly without the need for any stimulus. Voluntary blinking is that performed by the subject consciously. Reflex blinking, also named corneal reflex, is a rapid short-lived closing movement produced by various external stimuli, including bright lights [6], approaching objects, loud noises and corneal, conjunctival or eyelash rubbing. It is a reliable measure of afferent trigeminal VI and efferent facial nerve VII fibres [7]. Closure of stimulated eyelids is referred to as direct corneal reflex (ipsilateral), and closure of contralateral lids is termed the consensual corneal reflex. The consensual reflex is any reflex observed on one side of the body when the other side has been stimulated. This reflex is mainly evidenced in the process of pupil contraction of both eyes when only one of them is illuminated. Hence, in humans, these two responses are thought to be identical [8,9], and any divergence between them is attributed to neurological pathology. Nevertheless, some studies have found that the amplitude of the consensual response is smaller than that of the direct one [10,11], and that it is sex-dependent [12]. Regarding the analysis of corneal direct and consensual reflexes, in normal animals, the direct reflex is typically more pronounced than the consensual reflex [13].

The application of electrodes [14–21], the use of the magnetic search coil technique [22–25] and non-contact recording procedures, such as video oculography [26–30], are techniques that have been used to evaluate the eye blinking. Electrophysiological examinations provide the two blink reflex components (R1 and R2 responses) [21]. The R2 response is typically present bilaterally. In normal subjects, the R2 direct response to the stimulus is usually larger than the consensual one [13,20,31,32]. Yet despite the corneal reflex's diagnostic potential and reports of the direct reflex being more pronounced than the consensual reflex, we found no works in the literature that describe such a statement in depth and through a non-invasive image processing-based method. Only one previous communication by some of the authors of this paper includes indications of differences, but contain very few data [6].

The objective of this study was to analyse the differences in blinking kinematics of spontaneous and reflex blinks, distinguishing between direct and consensual reflexes, using for the first time a self-developed, non-invasive and image processing-based method. This work studied some dynamic characteristics of spontaneous blinking and the corneal reflex, and their significant differences. Blinking data are obtained by video sequences recorded with a high-speed camera at a rate of 240 frames per second (fps) and then processed off-line with Matlab (The MathWorks, Inc., Natick, MA, USA). The data of light diffused by eyelids are adjusted to an exponentially modified Gaussian (EMG) function. The EMG was introduced in chromatography [33] for describing peak shape because of its better formal data fitting than other skewed distributions, and its straightforwardly interpretable parameters. More recently, the EMG has been suggested to be applicable to cell biology [34], psychophysiology [35], physiology [36], physics [37], computer science [38] and blinking analyses [6]. The proposed form of the function proposed by Delley [39] is

$$f(t) = a + he^{-\frac{1}{2}\left(\frac{t-\mu}{\sigma}\right)^2}\frac{\sigma}{\tau}\sqrt{\frac{\pi}{2}}erfcx\left(\frac{1}{\sqrt{2}}\left(\frac{\sigma}{\tau} - \frac{t-\mu}{\sigma}\right)\right) \tag{1.1}$$

where $erfcx(t) = \exp(t^2)erfc(t)$ is a scaled complementary error function, $a$ is the independent term, $h$ is the amplitude of Gaussian, $\sigma$ is the standard deviation of the normal distribution, $\mu$ is the mean of the normal distribution and $\tau$ is the exponent decay parameter. These parameters can be related to dynamic characteristics of eye blinking, and are used to compare spontaneous blinks and corneal reflexes and to describe their differences.

Next, the methodology that we followed is described, including both the subjects participating in the study and the employed material. Briefly, two video sequences, including spontaneous and corneal reflex blinks, were recorded with 25 subjects and were then off-line processed. A statistical analysis is presented in the Results section, which lead us to draw the conclusions.

## 2. Methods

The students, faculty and staff of the department were recruited as participants. Twenty-five subjects aged 20–61 years (33 ± 14 years, 16 women and 9 men) participated in the study (figure 1). They did not have any history of medications or neurological, eye or eyelid disorders that would affect blinking. They were all informed about the nature and purpose of the study, and gave their informed

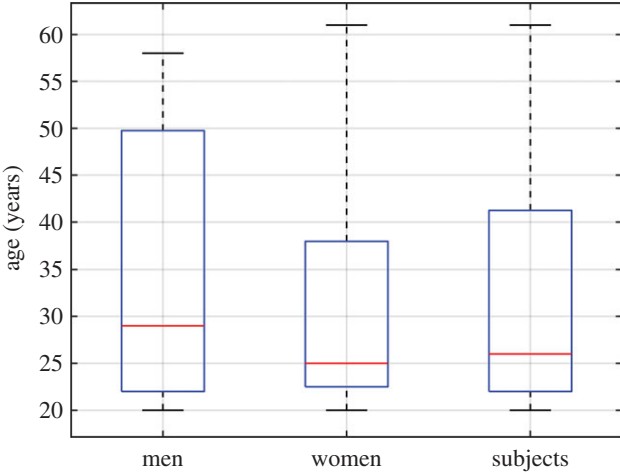

**Figure 1.** Boxplot of the age of the subjects participating in the study.

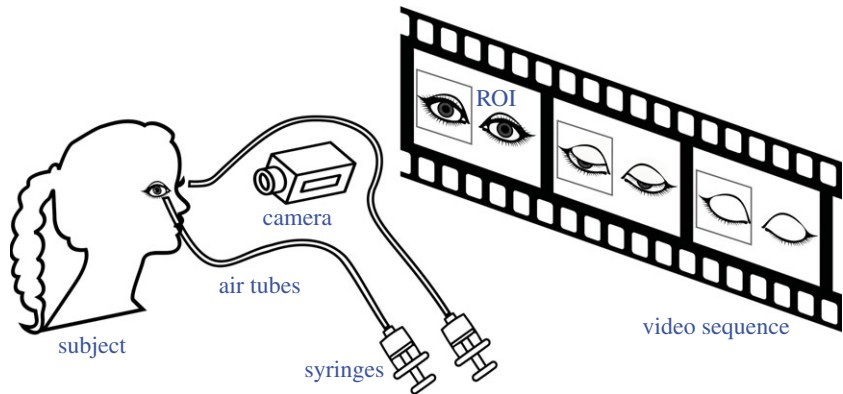

**Figure 2.** Scheme of the experimental set-up.

consent in writing. Additionally, all subjects of the study where pre-screened in order to discard subjects with excessive blinking or with an excessive number of incomplete blinks. We adhered to the Declaration of Helsinki principles and permission from the Ethical Committee of the University of Alicante was obtained (UA-2016-04-11).

The experimental set-up consisted of a chinrest on which the subjects rested their head and a high-speed commercial camera (GOPRO HERO 3+) working at 240 fps, which recorded both eyes during sequences lasting approximately 60 s [30,40,41]. While recording, the subjects were asked to blink naturally when they needed to and to not look away. Two air tubes connected to two syringes were pointed one at each eye. Syringes were randomly pressed, and a jet of air was shot without warning into one eye to stimulate the corneal reflex. The jet of air lasted less than 150 ms and air volume was around 20 ml. Two small pieces of paper were attached to the end of the tubes to detect when the jet of air was shot during video sequences. Two LED lamps (3500 K) were used to illuminate the subjects' eyes with an intensity of $1300 \pm 100$ lux. They were placed obliquely without interrupting the line of sight or dazzling the participants. Two video sequences for each participant spaced in time were recorded, in which spontaneous and reflex blinks were recorded. Figure 2 schematically represents the experimental set-up.

Videos were processed using Matlab. A rectangular region of interest (ROI) around each eye was first selected. This was done by hand in the first frame of each sequence to make the algorithm computationally lighter, but this selection was automatic in the following frames. The energy contained in each region was calculated in all the frames. The amount of light intensity diffused by the eye when it is open is almost constant. However, when the eyelid closes, the reflected light changes, and so does the intensity recorded by the camera. Therefore, blinks appear as a rapid increase and decrease in the light

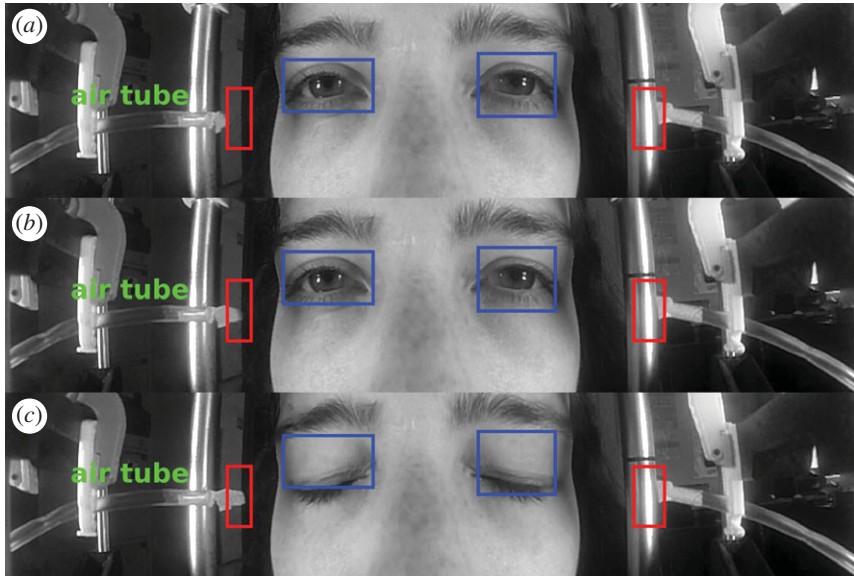

**Figure 3.** (*a*) Frame at an arbitrary moment of the sequence with eyelids open. (*b*) Frame at the instant when the jet of air is shot. (*c*) Frame when eyelids are closed during the corneal reflex. The ROIs selected to compute the light diffused by the eye are depicted by blue squares. The squared red areas are those selected to determine when the jet of air is shot and to which eye.

intensity recorded by the camera. This variation in intensity is directly related to changes in the position of eyelids. By finding these intensity peaks, we extracted blinks from video sequences [28].

Two other ROIs, including the pieces of paper at the end of the air tubes, were also selected by hand in the first frame and maintained for the other frames in the sequence. The energy in those ROIs was also tracked in time to determine when the jet of air was shot. In this way, we determined when corneal reflexes happened, and which ones were direct and consensual reflexes.

In figure 3, we represent three frames of one example sequence. We can see the air tubes at both sides of the face pointed at each eye. We squared in blue the example ROIs in which the intensity diffused by the eye was computed (the sum of the grey level of the pixel in the region), which provides the blink curve. In red, we boxed two example ROIs, used to detect the instant when the jet of air was shot. Frame (*a*) is taken before an air shot, and frame (*b*) is the frame at the instant when the jet of air was shot into the right eye. We can observe in the left red ROI how the piece of paper attached to the end of the air tube moves. In frame (*c*), the eyelid is closed, and the piece of paper has not yet returned to its initial position.

Blinks were extracted from the video sequences and sorted into spontaneous or reflexes. Data were reported separately for the left and right eye [24,42–44]. We considered corneal reflexes to be those blinks that immediately occurred after the jet of air was shot, and spontaneous blinks were the others. The next step consisted of least square fitting them to an EMG function (1.1) to gain the characteristic parameters ($a$, $h$, $\sigma$, $\mu$, $\tau$). Figure 4 provides an example of two blink signals and shows the fittings to the EMG function. Data correspond to both eyes during a corneal reflex. The instant when the jet of air was shot is marked by a black line. The data and fitted curve of the right eye are plotted in blue and those of the left one in red. Intensities are normalized prior to being fitted to the EMG function. The differences in the baseline value are irrelevant to our study and are discussed later.

As stated in the Introduction, the EMG is used because of its straightforwardly interpretable parameters. Here, $a$ and $h$ depend on the amount of intensity captured by the camera and do not characterize the blinks, $\mu$ merely provides information about the instant in the sequence when the blink occurs, while $\sigma$ and $\tau$ describe the shape of the blink peak. As seen in figure 4, the skewness of the blinking curves is positive, i.e. the tail is on the right. So $\sigma$ parameter describes the left side of the peak, the closing phase, whereas parameter $\tau$ refers to the right side, the opening. Note that none of them inform about the instants when blinks start or finish, but about the time taken to close and open. In general, some criteria could be stablished to determine the beginning and end of reflexes [40,41], for example, from the instant when the eyelid is closed. With corneal reflexes, we set the time to zero at the instant when the jet of air was shot. In this way, parameter $\mu$ was defined from the instant when the eye received the stimulus, and can also be used to compare direct and consensual corneal reflexes.

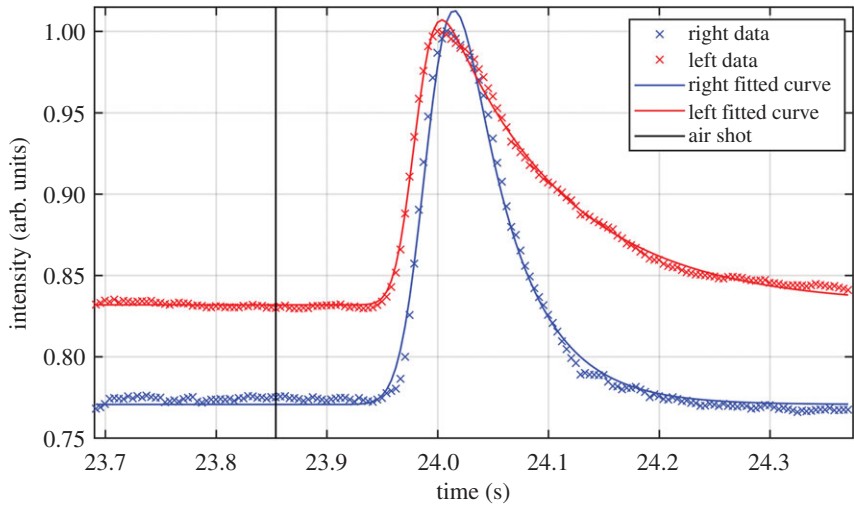

**Figure 4.** Data of a corneal reflex of the right (blue crosses) and left (red crosses) eyes. The blue and red lines are the curves that resulted from the fitting to (1.1) of the right and left eyes data, respectively. The instant when the jet of air was shot is marked by a black line.

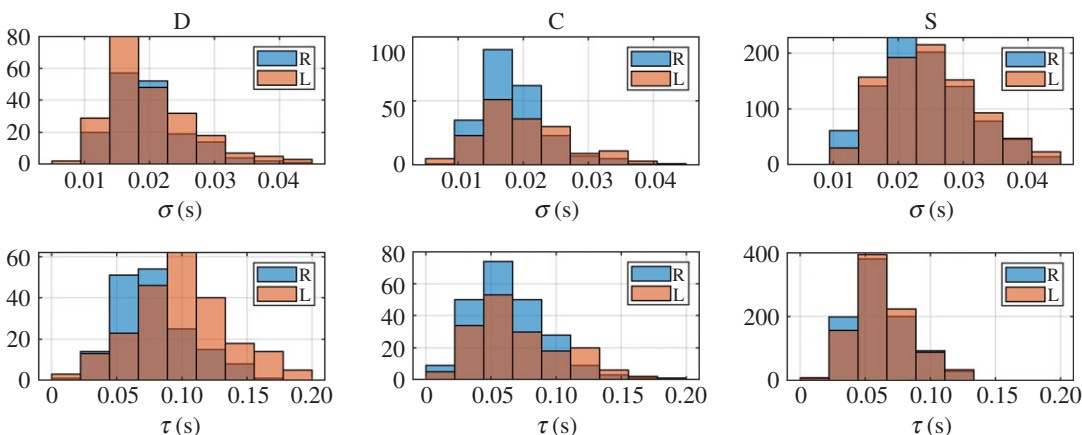

**Figure 5.** Histograms of parameters $\sigma$ and $\tau$ for the direct (D) and consensual (C) corneal reflexes, and the spontaneous blinks (S), of the right (R) and left (L) eyes.

## 3. Results

We processed the two sequences recorded per subject and classified the blinks as spontaneous blinks and corneal reflexes following the procedure explained above. We obtained 417 corneal reflexes and 933 spontaneous blinks. They were all fitted to the EMG function (1.1). The analysis of the differences in blinking kinematics of spontaneous and reflex blinks, distinguishing between direct and consensual reflexes, was based on the comparison of the characteristic parameters obtained from those fittings. We first analysed the statistics of the parameters of each blink type. In figures 5 and 6, we plotted the histograms of the parameters $\sigma$, $\tau$ and $\mu$ for the direct (D) and consensual (C) corneal reflexes, and the spontaneous blinks (S) of the right (R) and left (L) eyes.

In order to know how those independent variables (eye and blink type), in combination, affect any of the characteristic parameters obtained from the fittings to (1.1), we used a two-way ANOVA. For each subject a different number of blinks was obtained, so in order to get a balanced statistic analysis, we have constructed our database by randomly selecting three blinks from each type in each eye from every subject (3 blinks × 3 types × 2 eyes × 25 subjects). Therefore, each parameter dataset consists of a matrix with 75 rows (25 subjects × 3 blinks per subject) and six columns (2 eyes × 3 blink types per eye). The statistics for each characteristic parameter of these datasets are shown in figures 7–9, where we plotted the boxplots and histograms of each blink type for both eyes.

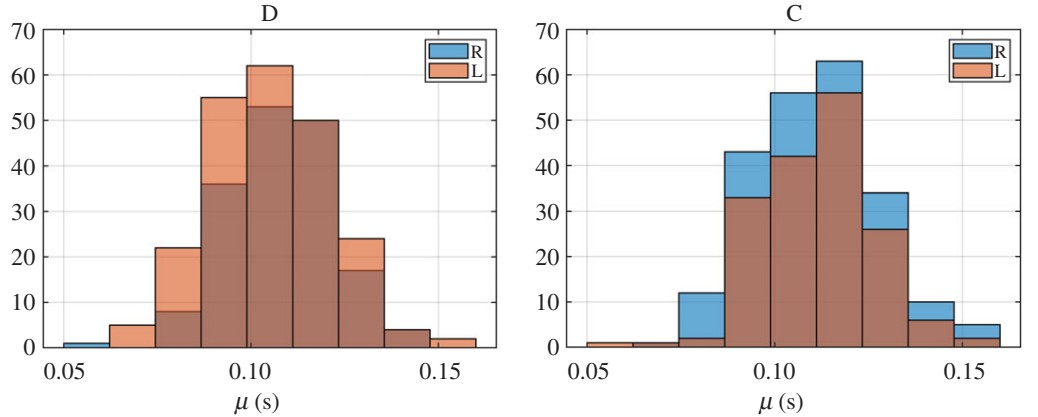

**Figure 6.** Histograms of parameter $\mu$, for the direct (D) and consensual (C) corneal reflexes, and the spontaneous blinks (S) of the right (R) and left (L) eyes.

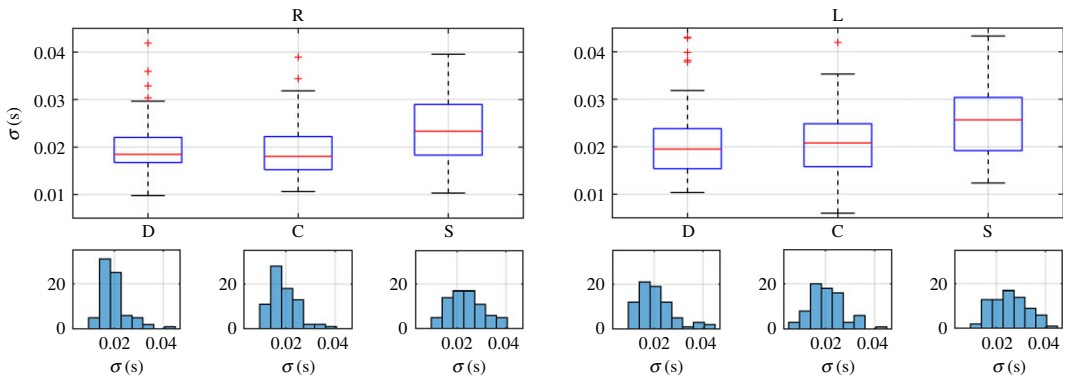

**Figure 7.** Boxplots and histograms of parameter $\sigma$ obtained for the right (R) and left (L) eyes and each blink type: direct (D) and consensual (C) corneal reflexes and spontaneous (S) blinks.

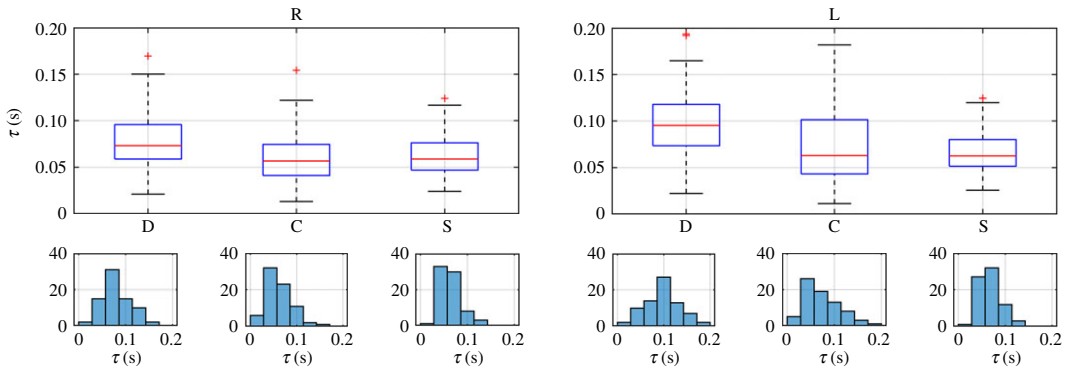

**Figure 8.** Boxplots and histograms of parameter $\tau$ obtained for the right (R) and left (L) eyes and each blink type: direct (D) and consensual (C) corneal reflexes and spontaneous (S) blinks.

All datasets of parameters meet the homoscedasticity necessary to use two-way ANOVA. The normality of each dataset was tested by the Shapiro–Wilk normality test because it has the most power for a given significance [45]. Neither the data from the parameter $\sigma$ nor those from parameter $\tau$ were normally distributed, as can be guessed from the histograms in figures 5, 7 and 8, but those from parameter $\mu$ and the transformations $\log_{10}(\sigma)$ and $\tau^{0.5}$ were. Therefore, when looking for statistically significant differences among the eyes and blink types, we performed a two-way ANOVA of these last parameters.

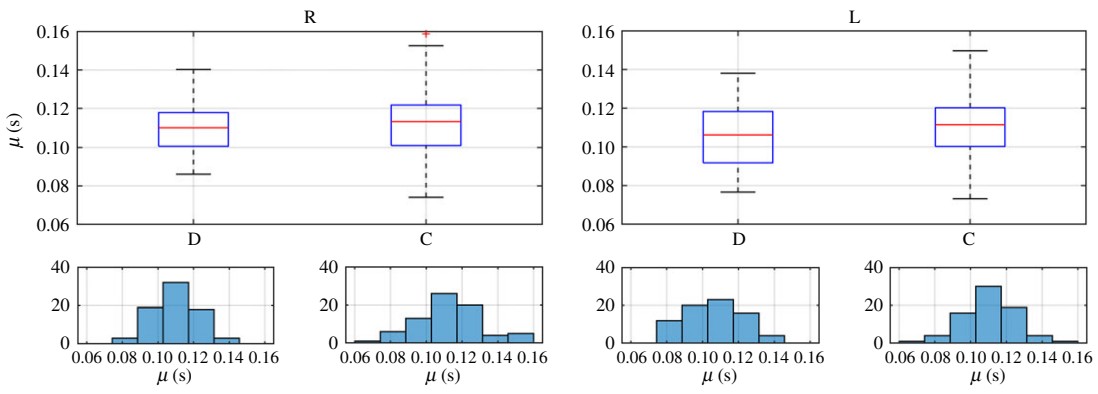

**Figure 9.** Boxplots and histograms of parameter $\mu$ obtained for the right (R) and left (L) eyes and the direct (D) and consensual (C) corneal reflexes.

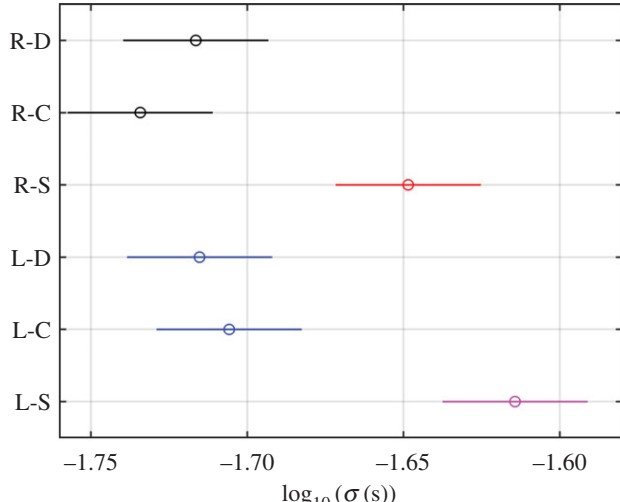

**Figure 10.** Multiple comparison test of the groups eye-blink type (R, right; L, left; D, direct; C, consensual; S, spontaneous) for $\log_{10}(\sigma)$.

**Table 1.** The ANOVA table for $\log_{10}(\sigma)$. SS, sum of squares; d.f., degrees of freedom; MS, mean square, defined by SS/d.f.; F, F-statistic value; p-value, p-value of the F-statistic value.

|  | SS | d.f. | MS | F | p-value |
|---|---|---|---|---|---|
| eye and blink type | 0.824 | 5 | 0.165 | 16.545 | $1.99 \times 10^{-14}$ |
| subjects | 1.629 | 24 | 0.068 | 6.813 | $1.84 \times 10^{-17}$ |
| interaction | 3.312 | 120 | 0.028 | 2.770 | $8.43 \times 10^{-13}$ |
| error | 2.989 | 300 | 0.010 |  |  |
| total | 8.755 | 449 |  |  |  |

Regarding the $\log_{10}(\sigma)$ parameter, the ANOVA result shown in table 1 establishes that it is influenced by the two factors and their interaction (p-values < 0.05). The significant interaction term hindered the ANOVA interpretation. Hence in order to evaluate, on the one hand, if there were differences between groups of eyes and blink types, and, on the other hand, if there were significant differences between subjects, a multiple comparison was made (figure 10).

The groups eye-blink type were R-D, R-C and R-S, corresponding to the direct, consensual and spontaneous reflex of the right eye, and L-D, L-C and L-S, corresponding to those of the left one, both, respectively. In figure 10, each group mean is represented by a symbol, and the mean squared

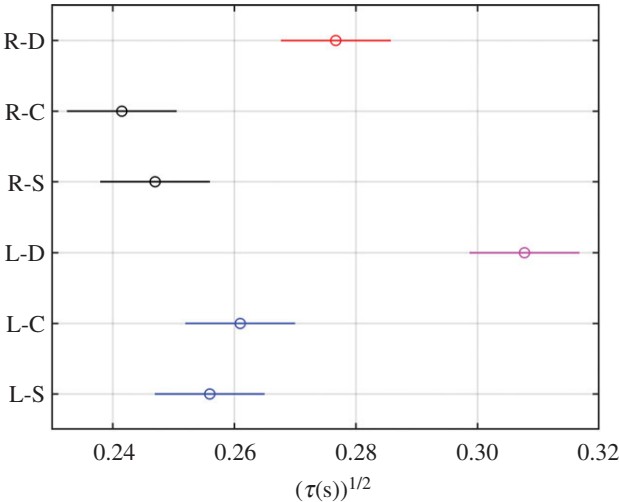

**Figure 11.** Multiple comparison test of the groups eye-blink type (R, right; L, left; D, direct; C, consensual; S, spontaneous) for $\tau^{0.5}$.

**Table 2.** The ANOVA table for $\tau^{0.5}$. SS, sum of squares; d.f., degrees of freedom; MS, mean square, defined by SS/d.f.; F, F-statistic value; p-value, p-value of the F-statistic value.

|  | SS | d.f. | MS | F | p-value |
|---|---|---|---|---|---|
| eye and blink type | 0.220 | 5 | 0.044 | 29.251 | $3.62 \times 10^{-24}$ |
| subjects | 0.450 | 24 | 0.019 | 12.441 | $1.70 \times 10^{-32}$ |
| interaction | 0.459 | 120 | 0.004 | 2.537 | $6.32 \times 10^{-11}$ |
| error | 0.452 | 300 | 0.002 |  |  |
| total | 1.582 | 449 |  |  |  |

**Table 3.** The ANOVA table for $\mu$. SS, sum of squares; d.f., degrees of freedom; MS, mean square; defined by SS/d.f.; F, F-statistic value; p-value, p-value of the F-statistic value.

|  | SS | d.f. | MS | F | p-value |
|---|---|---|---|---|---|
| eye and blink type | 0.0017 | 3 | 0.0006 | 3.9445 | $9.20 \times 10^{-3}$ |
| subjects | 0.0212 | 24 | 0.0009 | 6.0768 | $6.83 \times 10^{-14}$ |
| interaction | 0.0177 | 72 | 0.0002 | 1.6921 | $2.30 \times 10^{-3}$ |
| error | 0.0291 | 200 | 0.0001 |  |  |
| total | 0.0698 | 299 |  |  |  |

error is represented by a line extending from the symbol. Two group means were not significantly different if their intervals overlapped. In both eyes, there were no differences between the direct and consensual corneal reflexes, but spontaneous blink differed from them. Moreover, no differences appeared between eyes for any blink type, as was to be expected in subjects without pathologies and in line with other works in the literature [20].

The analysis of the parameter $\tau^{0.5}$ shown in table 2 also reports the existence of significant differences between eyes and blink types, between subjects and, again, their interaction (p-values < 0.05).

As with the analysis of the above parameter, the multiple comparisons plotted in figure 11 clarify what these differences were. In this case, the direct corneal reflex differed from the others in both eyes. The comparison made between eyes showed that both the direct and consensual corneal reflexes of the right eye had lower parameter values than those of the left eye, but, again, no significant differences appeared in spontaneous blinking.

The two-way ANOVA results of the parameter $\mu$ shown in table 3 indicate that groups eye-blink type, subjects and their interaction affected this parameter (p-values < 0.05). A multiple comparison test was

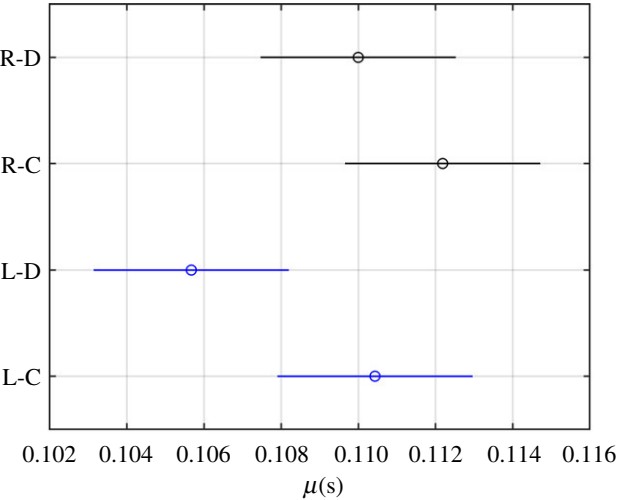

**Figure 12.** Multiple comparison test of the groups eye-blink type (R, right; L, left; D, direct; C, consensual) for $\mu$.

**Table 4.** Means and standard deviations of each parameter of each group eye-blink type (R, right; L, left; D, direct; C, consensual; S, spontaneous).

|  | R-D | R-C | R-S | L-D | L-C | L-S |
|---|---|---|---|---|---|---|
| $\sigma(s)$ | $0.020 \pm 0.006$ | $0.019 \pm 0.005$ | $0.024 \pm 0.007$ | $0.020 \pm 0.007$ | $0.021 \pm 0.007$ | $0.025 \pm 0.008$ |
| $\tau(s)$ | $0.08 \pm 0.03$ | $0.06 \pm 0.03$ | $0.06 \pm 0.02$ | $0.10 \pm 0.04$ | $0.07 \pm 0.04$ | $0.07 \pm 0.02$ |
| $\mu(s)$ | $0.110 \pm 0.012$ | $0.112 \pm 0.018$ |  | $0.106 \pm 0.016$ | $0.110 \pm 0.014$ |  |

run to clarify the differences between groups (figure 12), and we can see that there were not differences between the direct and consensual corneal reflexes in any eye and between eyes with any reflex type. The only significant difference lay in the $p$-value $< 0.05$ in the ANOVA between the consensual reflex of the right eye and the direct reflex of the left eye.

The lack of normality needed to apply a two-way ANOVA analysis to the distributions of the parameters $\sigma$ and $\tau$ was overcome by transforming them. However, the significant differences that have been found could be extended to the untransformed variables. Table 4 presents the summarized results of the mean and standard deviation of each parameter of each group eye-blink type.

The differences found in parameter $\sigma$ allowed us to conclude that the spontaneous blink took longer to close eyes than the corneal reflexes, while direct and consensual reflexes took the same time. Moreover, no significant differences were found between right and left eyes. The analysis of parameter $\tau$, which describes the duration of the opening phase, allowed us to conclude that the direct reflex blink took longer to open eyes than the consensual reflex and spontaneous blinks. Moreover, the comparison made between eyes revealed that both the direct and consensual corneal reflexes of the right eye had lower parameter values than those of the left eye, and there were no significant differences in spontaneous blinking. The differences between the right and the left eye could be due to the asymmetry of blinking [44], but it is likely that they were due to the asymmetries of the experimental set-up (the way subjects rested the faces, the arrangement of the air tubes, the force with which each syringe was pressed, etc.) because they appeared only in the corneal reflex. Finally, no differences appeared between the right and left eyes or between the direct and consensual reflexes in the parameter $\mu$.

## 4. Discussion and conclusion

Spontaneous blinks and corneal reflexes were assessed by distinguishing between right and left eyes using a self-developed, non-invasive and image processing-based method. Some parameters from the fitting to EMG curves were used to characterize and compare the blinks.

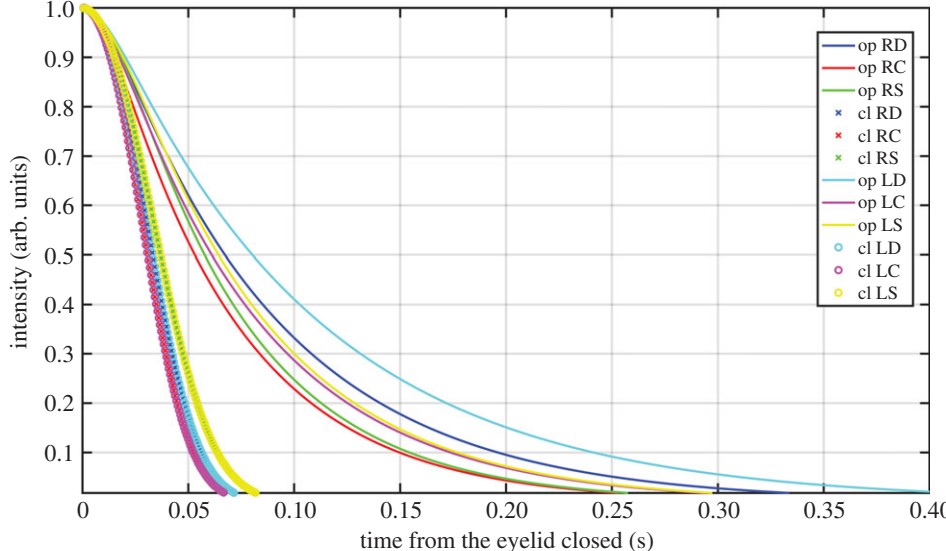

**Figure 13.** Intensity diffused in each phase (op, opening; cl, closing) for each eye (R, right; L, left) and blink type (D, direct; C, consensual; S, spontaneous) taking the closed eyelid as the origin.

In figure 13, we represent the variation of the intensity diffused in each phase for each eye and blink type, according to the model and by taking the closed eyelid as the origin. We have used the same parameter $\mu$ for all the types obtained as the average value of those in table 4. The eyelid was considered closed at the peak of the curve. All the intensity curves were normalized to 1, and the above conclusions were drawn. The closing phase lasted longer with the spontaneous blink, but with no differences between right and left eyes (green crosses and yellow ochre circles in figure 13). The opening phase was slower with the direct corneal reflex and differences between both eyes were found (light and dark blue lines). These results are consistent with those reported in previous works [13,20,31,32] that assert the direct reflex is typically more pronounced than the consensual reflex. Differences in closure can be understood as a defence mechanism: when the eyes are attacked, they react more quickly than normal. This explains how the spontaneous blink takes longer to close than the corneal reflexes. Regarding eye opening, the eye that suffers aggression remains closed longer in order to protect itself.

Ethics. We adhered to the Declaration of Helsinki principles and permission from the Ethical Committee of the University of Alicante was obtained (UA-2016-04-11).

Data accessibility. Data available from the Institutional Repository of the University of Alicante: http://rua.ua.es/dspace/handle/10045/107152.

Authors' contributions. J.E. carried out the laboratory work, participated in data analysis, participated in the design of the study and drafted the manuscript; J.P. carried out the statistical analyses and critically revised the manuscript; D.M. coordinated the study and helped draft the manuscript. All authors gave final approval for publication and agree to be held accountable for the work performed therein.

Competing interests. We declare we have no competing interests

Funding. We received no funding for this study.

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
