## [Reviewer comments · Royal Society Open Science]

Review History

RSOS-201016.R0 (Original submission)

Review form: Reviewer 1

Is the manuscript scientifically sound in its present form?

No

Are the interpretations and conclusions justified by the results?

No

Is the language acceptable?

Yes

Do you have any ethical concerns with this paper?

No

Have you any concerns about statistical analyses in this paper?

No

Recommendation?

Major revision is needed (please make suggestions in comments)

Comments to the Author(s)

In the paper entitled "Comparative analysis of spontaneous blinking and the corneal reflex" the authors investigated in 25 healthy subjects some dynamic characteristics of spontaneous blinking and reflex blinking, and their significant differences by using a video-oculography system, even if rationale is not established. They found that during the closing phase, direct and consensual responses of blink reflex are faster than spontaneous blinking, but there was no significant difference between them, nor between right and left eye. On the contrary, during the opening phase, the direct response was the slowest and significant differences appeared between right and left eye.

The statistical analysis is well performed, however some major concerns are present:

1. The rationale of research should be better explained in the Introduction and in the Abstract.
2. Since spontaneous blinking may be influenced by cognitive functions (Jongkees BJ and Colzato LS, 2016), I believe that the authors should have investigated these aspects in the participants enrolled in the study.
3. The authors enrolled 25 subjects aged 20-61 years (33 ± 14 years, 16 women and 9 men). Since blink reflex as well as spontaneous blinking may depend on age and sex (Peddireddy et al, 2006), the authors should have investigated any differences due to sex or age.
4. In the Methods the authors should give more information about the setting where subjects are studied (for example was it the same for all the participants? Was the lighting of the room artificial or not?).
5. To exclude subjects with increased spontaneous blinking the authors should have evaluate the spontaneous blink at rest.
6. I think the authors should better discuss their results.

Review form: Reviewer 2

Is the manuscript scientifically sound in its present form?

Yes

Are the interpretations and conclusions justified by the results?

Yes

Is the language acceptable?

Yes

Do you have any ethical concerns with this paper?

No

Have you any concerns about statistical analyses in this paper?

Yes

Recommendation?

Accept with minor revision (please list in comments)

Comments to the Author(s)

The paper investigates the blinking behaviour of two different types of blinking: spontaneous and corneal reflex (direct and consensual) in humans. The study used video recordings of eyes during blinking behaviour under natural and stimulation conditions (stimulation is performed randomly using air jets). Although the authors were motivated by the medical applications of characterizing the blinking behaviour, the data acquisition was performed from healthy subjects only. It would add more value to the paper if the data involved patients and control groups.

Comments:

- On page 3, line 48: Can you please provide more details about the difference between direct and consensual reflex?
- The authors used the amount of light reflected from the ROI to register eye blinking. I believe this technique is highly affected by ambient light and the size of the ROI. Can the authors try different techniques like eye alignment?
- On page 5, line 37: Did the author fix the ambient light intensity in all the recordings? Please, mention the light intensity value in lux (or lumens).
- What is the distance between the air tubes and the eyes?
- Are the syringes controlled manually? Does, the piece of paper used to detect the stimulus onset affect the air-flow?
- Why didn't the authors record more than two sequences (or longer sequences)? I believe the final number of blinks in each group is not sufficient to draw reliable conclusions.

Decision letter (RSOS-201016.R0)

Dear Dr Espinosa

The Editors assigned to your paper RSOS-201016 "Comparative analysis of spontaneous blinking and the corneal reflex" have now received comments from reviewers and would like you to revise the paper in accordance with the reviewer comments and any comments from the Editors. Please note this decision does not guarantee eventual acceptance.

Please submit your revised manuscript and required files (see below) no later than 21 days from today's (ie 20-Oct-2020) date. Note: the ScholarOne system will 'lock' if submission of the revision is attempted 21 or more days after the deadline. If you do not think you will be able to meet this deadline please contact the editorial office immediately.

on behalf of Professor Pietro Cicuta (Subject Editor)
openscience@royalsociety.org

Associate Editor Comments to Author:

The reviewers raise a number of critical comments in relation to your work. You need to carefully address their comments in a revision, and provide a point-by-point response to them. Please bear in mind that you will not generally be permitted to revise your paper further.

Reviewer comments to Author:

Reviewer: 1
Comments to the Author(s)

In the paper entitled "Comparative analysis of spontaneous blinking and the corneal reflex" the authors investigated in 25 healthy subjects some dynamic characteristics of spontaneous blinking and reflex blinking, and their significant differences by using a video-oculography system, even if rationale is not established. They found that during the closing phase, direct and consensual responses of blink reflex are faster than spontaneous blinking, but there was no significant difference between them, nor between right and left eye. On the contrary, during the opening phase, the direct response was the slowest and significant differences appeared between right and left eye.

The statistical analysis is well performed, however some major concerns are present:

1. The rationale of research should be better explained in the Introduction and in the Abstract.
2. Since spontaneous blinking may be influenced by cognitive functions (Jongkees BJ and Colzato LS, 2016), I believe that the authors should have investigated these aspects in the participants enrolled in the study.
3. The authors enrolled 25 subjects aged 20-61 years (33 ± 14 years, 16 women and 9 men). Since blink reflex as well as spontaneous blinking may depend on age and sex (Peddireddy et al, 2006), the authors should have investigated any differences due to sex or age.
4. In the Methods the authors should give more information about the setting where subjects are studied (for example was it the same for all the participants? Was the lighting of the room artificial or not?).
5. To exclude subjects with increased spontaneous blinking the authors should have evaluate the spontaneous blink at rest.
6. I think the authors should better discuss their results.

Reviewer: 2
Comments to the Author(s)

The paper investigates the blinking behaviour of two different types of blinking: spontaneous and corneal reflex (direct and consensual) in humans. The study used video recordings of eyes during blinking behaviour under natural and stimulation conditions (stimulation is performed randomly using air jets). Although the authors were motivated by the medical applications of characterizing the blinking behaviour, the data acquisition was performed from healthy subjects only. It would add more value to the paper if the data involved patients and control groups.

Comments:

- On page 3, line 48: Can you please provide more details about the difference between direct and consensual reflex?
- The authors used the amount of light reflected from the ROI to register eye blinking. I believe this technique is highly affected by ambient light and the size of the ROI. Can the authors try different techniques like eye alignment?
- On page 5, line 37: Did the author fix the ambient light intensity in all the recordings? Please, mention the light intensity value in lux (or lumens).
- What is the distance between the air tubes and the eyes?
- Are the syringes controlled manually? Does, the piece of paper used to detect the stimulus onset affect the air-flow?
- Why didn't the authors record more than two sequences (or longer sequences)? I believe the final number of blinks in each group is not sufficient to draw reliable conclusions.

===PREPARING YOUR MANUSCRIPT===

===PREPARING YOUR REVISION IN SCHOLARONE===

Author's Response to Decision Letter for (RSOS-201016.R0)

See Appendix A.

RSOS-201016.R1 (Revision)

Review form: Reviewer 1

Is the manuscript scientifically sound in its present form?

Yes

Are the interpretations and conclusions justified by the results?

Yes

Is the language acceptable?

Yes

Do you have any ethical concerns with this paper?

No

Have you any concerns about statistical analyses in this paper?

No

Recommendation?

Accept as is

Comments to the Author(s)

I thank the authors for the answers and I endorse the final version of the manuscript for publication

Review form: Reviewer 2

Is the manuscript scientifically sound in its present form?

Yes

Are the interpretations and conclusions justified by the results?

Yes

Is the language acceptable?

Yes

Do you have any ethical concerns with this paper?

No

Have you any concerns about statistical analyses in this paper?

No

Recommendation?

Accept as is

Comments to the Author(s)

The authors have addressed all comments. No additional comments.

Decision letter (RSOS-201016.R1)

Dear Dr Espinosa,

It is a pleasure to accept your manuscript entitled "Comparative analysis of spontaneous blinking and the corneal reflex" in its current form for publication in Royal Society Open Science.

on behalf of Prof Pietro Cicuta (Subject Editor)
openscience@royalsociety.org

Reviewer comments to Author:
Reviewer: 1

Comments to the Author(s)
I thank the authors for the answers and I endorse the final version of the manuscript for publication

Reviewer: 2

Comments to the Author(s)
The authors have addressed all comments. No additional comments.

Appendix A

Reviewer: 1

In the paper entitled "Comparative analysis of spontaneous blinking and the corneal reflex" the authors investigated in 25 healthy subjects some dynamic characteristics of spontaneous blinking and reflex blinking, and their significant differences by using a video-oculography system, even if rationale is not established. They found that during the closing phase, direct and consensual responses of blink reflex are faster than spontaneous blinking, but there was no significant difference between them, nor between right and left eye. On the contrary, during the opening phase, the direct response was the slowest and significant differences appeared between right and left eye.

The statistical analysis is well performed, however some major concerns are present:

1. The rationale of research should be better explained in the Introduction and in the Abstract.

We would like to thank Reviewer's suggestion. The reasons of research were not clearly stated in the manuscript.

Noninvasive measurement of blinking has been a research subject by the authors of the manuscript since 2010 [26]. The implementation of image processing algorithms allows that the contact-free recording of the annexes of the eye can be used to gather and analyze quantitative blink kinematic parameters precisely, with an accuracy comparable to, or even better than, the search coil method. This fact together with the central nervous system connection with blinking encouraged us to deep in its research during the last years. In 2015, we presented a high-resolution binocular video-oculography system aimed to the assessment of pupillary light reflex, with which we detected an early incomplete blink and an upward eye movement [Espinosa et al. "A high-resolution binocular video-oculography system: assessment of pupillary light reflex and detection of an early incomplete blink and an upward eye movement," *Bimed. Eng. Online* 2015]. Corneal reflex, like the pupillary one, can be elicited by a light stimulus but we realized the few works that had referred to the study of the direct and consensual corneal reflexes considering their importance for example as indicator of both neurological and optic nerve pathologies. We preliminary addressed the assessment of direct and consensual corneal reflex and spontaneous blink in SPIE. Result suggested there existed differences between both reflexes, but statistic was poor. In this work, we modified the experimental setup, changing light stimulus by air jet shot, we evaluated a larger population and performed a complete statistical analysis. The objective of this study was to analyze the differences in blinking kinematics of spontaneous and reflex blinks, distinguishing between direct and consensual reflexes, using for the first time a self-developed, non-invasive, and image processing-based method.

2. Since spontaneous blinking may be influenced by cognitive functions (Jongkees BJ and Colzato LS, 2016), I believe that the authors should have investigated these aspects in the participants enrolled in the study.

3. The authors enrolled 25 subjects aged 20-61 years (33 ± 14 years, 16 women and 9 men). Since blink reflex as well as spontaneous blinking may depend on age and sex (Peddireddy et al, 2006), the authors should have investigated any differences due to sex or age.

The study population includes men and women within a wide adult range and no specific cognitive function was considered when making the measurements. Our interest is not in determining differences between age or sex but between types of blinking (spontaneous and direct and consensual reflexes). For this purpose, we proposed a technique that allows to extract kinematic parameters from isolated blink curves that are directly related to the opening and closing phases, but no rate analysis was performed. The central dopamine function is closely associated with the spontaneous blink rate and Jongkees BJ and Colzato LS, 2016 provides a comprehensive review. On the other hand, Peddireddy et al, 2006 hypothesized that blink reflex may be influenced by the effects of aging and gender; and with that aim

evaluated electromyographic responses of two groups of 15 men of different mean age and other two ones of women.

We do not doubt about the interest of applying our technique to such analyses, but both are out of the scope of the manuscript and do not affect the validity of our results.

4. In the Methods the authors should give more information about the setting where subjects are studied (for example was it the same for all the participants? Was the lighting of the room artificial or not?).

Two LED lamps (3500 K) were used to illuminate the subjects' eyes with an intensity of 1300 ± 100 lux. They were placed obliquely without interrupting the line of sight or dazzling the participants. We have included that information in the manuscript to better describe the experimental setup. Responding to the issues raised by the reviewer, the lighting was artificial, and the illumination was the same for all the participants.

5. To exclude subjects with increased spontaneous blinking the authors should have evaluate the spontaneous blink at rest.

We agree with reviewer 1 that one way to stablish a criterium to exclude subjects with increased spontaneous blinking is to evaluate the blink at rest. All subjects of the study where pre-examined in order to discard subjects with excessive blinking or with an excessive number of incomplete blinks. The participants in this study were selected from a group of people that already participated in previous related studies. Therefore, only those with normal blinks and with no tendency to incomplete blinks were selected. We recognize that this was not a proper selection, but also the sampling was not completely blind. The measurements were performed without asking the participants making any work (reading, following a target, ...) but looking at a distance point. In any case, the blink rate is out of the scope of this work; we are interested in isolated blinks and the establishment of characteristic parameters.

We have added a sentence in the text recognizing that a pre-evaluation of the subjects was performed in order to avoid people with excessive number of blinks or incomplete blinks.

6. I think the authors should better discuss their results.

We appreciate the reviewer's suggestion and have extended the discussion of the results.

Reviewer: 2

The paper investigates the blinking behaviour of two different types of blinking: spontaneous and corneal reflex (direct and consensual) in humans. The study used video recordings of eyes during blinking behaviour under natural and stimulation conditions (stimulation is performed randomly using air jets). Although the authors were motivated by the medical applications of characterizing the blinking behaviour, the data acquisition was performed from healthy subjects only. It would add more value to the paper if the data involved patients and control groups.

Comments:

- On page 3, line 48: Can you please provide more details about the difference between direct and consensual reflex?

We have modified the manuscript presenting extended information about the direct and consensual reflex, according reviewer's suggestion. The consensual reflex is any reflex observed on one side of the body when the other side has been stimulated. This reflex is mainly evidenced in the process of pupil contraction of both eyes when only one of them is illuminated. Hence, in humans, these two responses are thought to be identical[8,9], and any divergence between them is attributed to neurological pathology. Nevertheless, some studies have found that the amplitude of the consensual response is smaller than that of the direct one[10,11], and that it is sex-dependent[12]. Regarding the analysis of corneal direct and consensual reflexes, in normal animals, the direct reflex is typically more pronounced than the consensual reflex[13].

- The authors used the amount of light reflected from the ROI to register eye blinking. I believe this technique is highly affected by ambient light and the size of the ROI. Can the authors try different techniques like eye alignment?

The technique is based in quantifying the change of the light diffused in a region of the eye (ROI). That ROI will be crossed by the eyelid when blinking and the registered intensity will change. The intensity obtained at each instant has no physical sense but its variation in time is directly related to the position of the eyelid during blinking. All analyzed parameters (μ , σ and τ) have time dimensions so they are not affected by the intensity of light. Moreover, they are obtained after normalizing the curves to the maximum intensity (see figure 4). Hence, the technique is not affected by ambient light or the size of the ROI if they do not vary in the sequence interval and the ROI does not include other element that change the diffused light during that interval.

Regarding the suggestion of trying eye alignment, we doubt it could be used to reach our results. In any case, the performance of technique has been broadly validated in our published papers an also from other authors and we do not consider necessary employing another one to compare the results.

- On page 5, line 37: Did the author fix the ambient light intensity in all the recordings? Please, mention the light intensity value in lux (or lumens).

Two led lamps (3500 K) were used to illuminate the subjects' eyes with an intensity of 1300 ± 100 lux. They were placed obliquely without interrupting the line of sight or dazzling the participants. We have included that information in the manuscript to better describe the experimental setup.

- What is the distance between the air tubes and the eyes?

Tubes were place at 3 cm from the eyes. We have included that information in the manuscript to better describe the experimental setup.

- Are the syringes controlled manually? Does, the piece of paper used to detect the stimulus onset affect the air-flow?

Syringes were controlled manually. The piece of paper used to detect the stimulus was a tissue lightweight paper of approximately 10 mm x 5 mm. It probably affected the airflow a little but not enough to avoid the stimulus of the corneal reflex, which was our interest.

- Why didn't the authors record more than two sequences (or longer sequences)? I believe the final number of blinks in each group is not sufficient to draw reliable conclusions.

Longer sequences could have affected the fatigue of participants. 60-seconds sequence is enough to detect 9 spontaneous blinks and stimulate 4 reflex blinks (in average). Moreover, it easy to find recent works with similar length records (Sanchis-Jurado et al. Non-invasive high-speed blinking kinematics characterization, 2020). With those average values, we got around 18 spontaneous and 8 reflex blinks per eye and subject from that we randomly selected three of each type. To perform a two-way ANOVA of the characterizing parameters of blinking it is enough with 3 measures per parameter and subject.